# Simplified Artificial Intelligence Terminology for Pathologists

**DOI:** 10.3390/diagnostics15131699

**Published:** 2025-07-03

**Authors:** Fatemeh Zabihollahy, Michael Mankaruos, Maxim Mohareb, Timothy Youssef, Yasaman Soleymani, George M. Yousef

**Affiliations:** 1Laboratory Medicine Program, University Health Network, Toronto, ON M5G 2C4, Canada; fatemeh.zabihollahy@uhn.ca (F.Z.); michaelmankaruos@gmail.com (M.M.); maximmohareb@gmail.com (M.M.); timothyyoussef2006@gmail.com (T.Y.); soleymani.yasaman04@gmail.com (Y.S.); 2Department of Laboratory Medicine and Pathobiology, University of Toronto, Toronto, ON M5G 2C4, Canada

**Keywords:** Artificial intelligence, computer-aided diagnosis, deep learning, digital pathology, machine learning, foundation models

## Abstract

The expanding shift towards digital pathology in clinical practice globally highlights its potential to enhance patient care through artificial intelligence (AI)-powered, computer-assisted diagnostics. Effective communication between AI scientists and pathologists is crucial for this transformation, though their differing technical languages can pose challenges. The manuscript aims to offer simplified explanations of common AI terminology, along with practical examples and illustrations, to help pathologists better grasp AI concepts. This review is divided into the following sections: AI technologies and algorithms in computational pathology; frameworks for training AI models; nomenclature of image analysis; and public datasets for computational pathology research. These sections collectively provide a comprehensive understanding of the current landscape and resources in computational pathology. The manuscript fosters better communication between these fields and showcases the advantages of AI technologies in pathology.

## 1. Introduction

There is a growing shift toward digital pathology in clinical practice worldwide [1]. In addition to the multiple benefits of digital pathology for clinical practice, the holy grail of digital pathology is that it serves as an enabler for artificial intelligence (AI) and utilizes computer-assisted diagnostics to improve the quality of patient care [2]. A major requirement in this regard is the need for active communication between AI scientists and pathologists. Oftentimes, these two groups speak different languages. The jargon used by AI scientists is sometimes confusing for pathologists and too technical to understand. On the other hand, AI scientists are not familiar with the details of clinical operations and standard image production for glass slides or scanned images in pathology.

In this manuscript, we try to bridge the gap between the two worlds by providing a simplified explanation of the different terminologies commonly used in the AI field. Although strict technical definitions are widely available, our efforts are focused on providing simplified terminology that will facilitate communication between pathologists and AI scientists. We also aim to provide reasonable illustrations and focus on practical examples that show the benefits of different AI technologies when applied to pathology. This article is not meant to be a glossary of terminology, but rather a simplified description that helps non-AI scientists understand AI concepts in a simplified way.

### AI in Pathology

AI is a vast field within computer science that empowers machines to perform tasks typically associated with human intelligence, such as image recognition, prediction and decision-making, and natural language understanding. AI has been extensively applied in pathology literature for the examination of tissues, cells, and organs, demonstrating its potential to become an integral part of the pathology workflow for diagnosing cancer and complex diseases. A remarkable survey, which gathered insights from 75 computational pathology experts from academia and industry, identified the most promising applications of AI in pathology as tissue segmentation, cell detection, mutation prediction, and treatment response prediction. The survey results indicated that predicting treatment response and survival directly from routine pathology slides, as well as predicting genetic alterations and gene expression, are seen as the most promising future applications [3].

As summarized in Table 1, the applications of AI in pathology can be categorized into three main areas, as detailed below.

Image analytics: AI enhances the clinical workflow by saving time and improving the accuracy of pathology image analysis. Example tasks include counting mitotic figures [4]; quantifying immunohistochemical (IHC) markers such as p53 [5], PDL1, estrogen receptor (ER), and HER2 [6]; and measuring tumor size [7].

Disease diagnostics: AI assists in identifying lesional tissue on histopathology slides. For example, AI can diagnose prostate cancer and predict the Gleason score in prostate core biopsies [8]. It can also be used to find perineural invasion [9], lymphovascular invasion [10,11], and lymph node metastasis in cancer [12,13,14]. It should be noted that the accuracy of these algorithms is not adequate to replace pathologists, and in most cases, they are used as a guide to save pathologists’ time, while the final decision is left for the pathologist.

Outcome prediction: Developing algorithms to predict patient outcomes is perhaps the most challenging task [15,16,17]. It involves correlating clinical information (oftentimes a combination of pathology image features [18], molecular changes reflected in pathology images [19], and other clinical and genomic parameters) to predict disease prognosis or therapy response. Additionally, histopathology AI analysis can be used to triage cases for molecular testing [20]. The term “integrated diagnostics” was recently introduced to emphasize the value of compiling multiple levels of information [21,22,23].

It should be noted that AI is not intended to replace pathologists, but rather to assist them in improving diagnostic quality. The term computer-assisted diagnostics has been recently introduced to emphasize this fact.

To facilitate the discussion, we have divided our review into the following sections: (1) different types of AI technologies and algorithms in computational pathology; (2) common frameworks for training AI models; (3) the nomenclature of image analysis; and (4) an overview of public datasets available for computational pathology research.

## 2. AI Technologies and Algorithms

An “algorithm” is a set of step-by-step instructions that are followed to perform a task or solve a problem. Algorithms are used in areas such as mathematics, computer programming, and AI. In computer programming, algorithms are used to perform calculations, search databases for information, and more. In AI, they are used to program computers to learn how to operate autonomously.

Different AI technologies and algorithms, including machine learning, neural networks, deep learning, convolutional neural networks, and generative AI, can be utilized for pathology image analysis, as detailed below.

### 2.1. Machine Learning and Neural Network Applications in Pathology

Machine learning, a subfield of artificial intelligence, excels at identifying patterns in data to make predictions without explicit human programming. Common machine learning algorithms include logistic regression, decision trees, random forests, support vector machines, K-nearest neighbors, and gradient boosting machines. In pathology, these algorithms are often applied to manually extracted quantitative features from digitized histopathology images, a technique known as pathomic analysis. Recent studies have used machine learning models to predict pathological complete response and disease-free survival in rectal cancer [24], assist in the diagnosis and prognosis of bladder cancer [25], and identify prognostic markers in clear cell renal cell carcinoma [26]. Deep learning, a specialized subset of machine learning, relies on neural networks, algorithms inspired by the structure of the human brain. A neural network is composed of interconnected layers (input, hidden, and output) that process complex data such as digital slides or pathology reports, enabling tasks like cancer grading, metastasis detection, and survival prediction.

### 2.2. Deep Learning and Convolutional Neural Network Applications in Pathology

Deep learning, a subset of machine learning, uses deep neural networks with multiple layers to perform complex tasks that require hierarchical data representation. These models can learn intricate patterns from large datasets. Ahmed et al. provide a comprehensive overview of the current and potential applications of both deep learning and machine learning in tumor pathology [27]. Key deep learning architectures include convolutional neural networks (CNNs), recurrent neural networks, generative adversarial networks, transformers, and autoencoders. CNNs, in particular, are specialized for visual data such as images and videos and use convolutional layers to automatically extract features such as edges, colors, and textures. This automatic feature extraction distinguishes CNNs from traditional neural networks, which require manual feature input, making CNNs especially effective for image analysis in pathology (Figure 1).

It is noteworthy that neural networks do not, by definition, require manual extraction of features. However, applying a traditional neural network directly to image data, without manual feature extraction, presents several limitations. First, traditional neural networks treat each pixel as an independent feature and ignore the spatial relationships that are critical for understanding visual patterns. They fail to capture hierarchical features and generalize poorly to variations in scale, position, or rotation within images. Additionally, this approach results in a massive number of parameters due to the high dimensionality of image inputs, making the model computationally expensive and prone to overfitting. As a result, while they may work for small or simple datasets, traditional neural networks are generally inefficient and ineffective for real-world image analysis compared to convolutional neural networks.

Abdelsamea et al. provided a comprehensive survey of convolutional neural network applications in histopathology images, covering tasks such as image classification, segmentation, localization, and detection [28].

### 2.3. Generative AI

Generative AI refers to a type of AI that creates new content, such as text, images, and music, by learning patterns from different existing sets of data (e.g., drawing a picture of a garden from a written description). Unlike traditional AI, which classifies or predicts based on input data, generative AI can produce original outputs that mimic the style and structure of the data it was trained on. Rashidi et al. discussed the transformative potential of generative AI models and explored their applications in pathology and medicine [29]. One of the most promising applications of generative AI is creating synthetic yet realistic, high-quality pathology images for training and validating AI models, helping overcome the challenge of limited annotated data [30,31].

## 3. Common Frameworks for Training AI Models

There are different approaches for training AI models, including supervised learning, unsupervised learning, weakly supervised learning, multiple-instance learning, self-supervised learning, transfer learning, and federated learning. Below, we will discuss these approaches briefly.

### 3.1. Supervised Learning

Supervised learning is a common framework used to train AI algorithms. It involves training a model on labeled (annotated) data, where the desired output is known. The model learns to map parts of the test image based on these examples [32]. Supervised learning methods construct AI models by learning from a large number of training examples, each with a label indicating its correct output (ground truth output). For instance, a model might be trained to distinguish between negative and positive metastatic pelvic lymph nodes in prostate cancer; in this case, the model would be given thousands of digital slides with their correct diagnoses to learn from. This strategy is widely used in computational pathology applications. For example, supervised learning was employed to predict the Gleason Score from 11,000 full diagnostic biopsy images [8]. Another example is an AI model trained in a supervised manner using more than 8000 labeled hematoxylin and eosin (H&E) digital slides of pelvic lymph nodes to detect and segment metastatic cancer cells [33].

### 3.2. Unsupervised Learning

Unsupervised learning models are trained using inputs that do not have associated outputs (no annotation or known outcomes). These models are tasked with grouping or clustering the input data in meaningful ways without prior knowledge of the correct output. Arevalo et al. proposed an unsupervised learning method for the automatic detection of basal cell carcinoma in histopathology images [34].

### 3.3. Weakly Supervised Learning

Although current supervised learning techniques have achieved great success, acquiring fully ground truth labels requires considerable time, money, and effort. Therefore, training AI models with weak supervision is highly desirable, particularly in computational pathology, where there are many target applications and often limited labeled training samples. Campanella et al. employed weakly supervised learning to address the challenge of insufficient labeled data for training an AI model on whole slide images [35].

In weakly supervised learning, the model is trained with incomplete, inexact, or imprecise labels associated with datasets [36]. Consider H&E image categorization, in which ground truth labels are diagnoses provided by pathologists. To design an AI in a weakly supervised manner with incomplete supervision, a large number of digital slides can be accessed from hospitals, while only a small subset of slides is annotated due to the high cost of human experts. Weakly supervised learning has different models. In “inexact supervision”, usually, we only have “image-level” labels rather than “object-level” labels. For instance, in identifying lymph nodes positive for cancer, a slide may be labeled as positive for metastatic cells, but the exact location of the metastases on the pathology slide is not specified. In “inaccurate supervision”, the given labels are not always ground truth (e.g., erroneous diagnoses), which can occur when the annotator is inexperienced or when some pathology slides are difficult to categorize.

### 3.4. Multiple-Instance Learning

Multiple-instance learning is a form of weakly supervised learning where training data is organized into subsets referred to as “bags”. In computational pathology, a bag may contain *patches extracted from one digital slide.* The bag is labeled as positive if at least one patch is positive, and negative if all instances are negative. The goal is to train an AI model to classify new bags correctly, either by identifying positive instances within positive bags or by distinguishing between positive and negative bags.

The multiple-instance learning approach uses reported diagnoses as labels for training, thereby avoiding expensive and time-consuming pixel-level manual annotations (i.e., having to annotate every leasional area on the entire slide). Campanella et al. employed multiple-instance learning for prostate cancer, basal cell carcinoma, and breast cancer metastases to axillary lymph nodes, demonstrating that this system has the ability to train accurate classification models at an unprecedented scale [35].

### 3.5. Self-Supervised Learning

Self-supervised learning is a type of unsupervised machine learning that leverages the data itself to generate labels, rather than relying on manually labeled data. In self-supervised learning, the model learns to predict parts of the data from other parts of the same data. This approach is particularly useful for tasks where labeled data is scarce or expensive to obtain. Recently, Zimmermann et al. adapted self-supervised learning for computational pathology, demonstrating its value on several benchmarks [37].

The differences between the aforementioned deep learning frameworks are illustrated in Figure 2. To clarify their distinctions, consider the example of training an AI model to detect lymph node metastasis in prostate cancer (PCa) using input data, such as hematoxylin and eosin (H&E) stained whole-slide images. In this context, the term “label” refers to the ground truth diagnosis, typically determined by experienced pathologists.

In supervised learning, the model is trained using both pixel-level annotations of cancerous and non-cancerous regions, as well as corresponding labels. This approach requires extensive manual annotation by expert pathologists. In contrast, unsupervised learning does not rely on any labels or annotations. Instead, the model identifies patterns within the data by clustering image regions based on feature similarity, with the hope that these clusters will correspond to biologically meaningful groups, such as malignant and benign areas.

Weakly supervised learning operates with partial annotations and labels. For instance, only some cancerous regions within the lymph node may be contoured, resulting in incomplete spatial annotation. This framework reduces the annotation burden while still leveraging some supervision. In multiple-instance learning, the model receives only slide-level labels indicating the presence or absence of metastasis without specific localization. The AI must learn to associate regions with labels indirectly, identifying the most informative instances within the slide.

Self-supervised learning shares the absence of labels and annotations with unsupervised learning, but introduces auxiliary tasks that generate pseudo-labels. These tasks are not directly related to the primary objective but help the model learn relevant features. For example, the model may be trained to predict the rotation angle (e.g., 90° or 180°) of an H&E image. Although predicting image orientation is unrelated to detecting lymph node metastasis, this task enables the model to develop a structural understanding of prostate tissue architecture. Consequently, self-supervised learning can substantially reduce the need for annotated data in downstream tasks such as metastasis detection.

### 3.6. Transfer Learning

Transfer learning is a useful machine learning technique that uses knowledge learned by an AI model from one task/data to a similar task/data. For instance, an AI model that has been trained to diagnose breast cancer can be fine-tuned using small labeled data for grading bladder cancer (Figure 3).

Adapting the transfer learning approach significantly reduces the amount of labeled data required to train an AI model for optimal performance. By minimizing the dataset size, transfer learning also alleviates computational costs associated with processing large datasets. Moreover, transfer learning enhances a model’s generalizability by involving fine-tuning (i.e., retraining) an existing model with a new dataset, thereby incorporating knowledge from multiple datasets. Transfer learning has been widely used in the literature for analyzing pathology data [38,39,40,41].

Although transfer learning is a useful technique in AI, the pre-trained model may not effectively transfer knowledge when the data distribution differs significantly between the source and target domains, resulting in lower-than-expected performance. As the majority of pre-trained models in the computer vision domain are trained on ImageNet, using those models may not add much value for analyzing pathology images.

Li et al. discussed how much off-the-shelf knowledge is transferable from natural images to pathology images [42]. Recently, some general-purpose deep learning models, referred to as “foundation models,” have been trained using a large number of pathology slides and can be fine-tuned for downstream tasks. A couple of foundation models in the field of pathology include Virchow, GigaPath, UNI, and CONCH [43,44,45,46]. Neidlinger et al. benchmarked histopathology foundation models on certain patient cohorts with various cancer types and demonstrated that models trained on distinct cohorts learn complementary features, outperforming state-of-the-art AI methods [47].

### 3.7. Federated Learning

Federated learning is a subset of machine learning techniques in which an *AI model is trained collaboratively* by multiple institutions. In this setting, a trained AI model is shared between the centers to be trained using their data, rather than sharing the data, which addresses potential patient privacy concerns and data access limitations in healthcare (Figure 4). Centers can adopt various learning approaches for training AI models, including supervised, unsupervised, weakly supervised, multiple-instance, and self-supervised learning. Lu MY. et al. demonstrated the usefulness of using federated learning as part of a larger algorithm for gigapixel whole slide images [48].

## 4. Nomenclature of Image Analysis

Computer vision is a field of AI that enables machines to interpret and make decisions based on visual data, much like the human visual system. In computer vision, computers are trained to “see” and understand the content of digital images and videos. Within computational pathology, computer vision techniques assist pathologists in diagnosing diseases by analyzing histopathology images through image classification, segmentation, and object detection.

### 4.1. Image Classification

Image classification is a process in computer vision that involves categorizing images based on specific characteristics (e.g., cancer vs. normal) (Figure 5). To determine these characteristics (rules), computers are trained to recognize and identify patterns in images that are useful for classification. For example, a computational approach using convolutional neural networks was developed to categorize breast cancers using histology images [49].

### 4.2. Image Segmentation

As shown in Figure 6, image segmentation is dividing an image into segments, or regions (e.g., normal, cancer, stroma), to simplify the representation of an image into something more meaningful and easier to analyze. Image segmentation is widely used for histopathology images for a variety of tasks, like tissue segmentation [50], nucleus identification [51], and tumor detection. For example, a recent study analyzed the complex kidney system, which comprises various components across multiple levels, including regions (cortex, medulla), functional units (glomeruli, tubules), and cells (podocytes, mesangial cells in the glomerulus) [52].

### 4.3. The Gold Standard and Ground Truth

The “gold standard” in machine learning and deep learning refers to the most accurate information available for the training and evaluation of an AI model. However, establishing this gold standard in the field of biomedical imaging and cancer diagnosis is particularly challenging and costly, as it requires extensive follow-up, pathologic diagnoses, molecular testing, and clinical outcomes. As a result, the term “ground truth” is often used as a surrogate for the gold standard. Ground truth data is assumed to be correct, based on direct observation or the expert opinion of experienced clinicians. Given the modest observer variability in interpreting pathology images [53], ground truth preparation often involves engaging experienced pathologists to analyze a given study. To ensure objectivity, this process may include double-checking annotations. Another approach is to have several pathologists analyze the same dataset, using either average statistics or a majority vote system to establish the ground truth.

## 5. Datasets in Computational Pathology

In machine learning and deep learning, a dataset refers to a collection of data used for training and testing AI models. In the field of pathology, datasets play a critical role in developing, training, validating, and benchmarking AI and machine learning models. These datasets generally fall into several broad categories based on their content and use. WSI datasets comprise digitized histopathology slides, often accompanied by metadata or clinical labels such as diagnosis, tumor grade, or biomarker status. Multi-modal datasets combine WSIs with complementary data types, such as genomics, transcriptomics, or radiology images, to enable integrative or “multi-omics” analysis. Temporal or longitudinal datasets track disease progression or treatment response over time, supporting prognostic modeling.

Dividing a dataset into subsets is known as “data splitting”. Typically, datasets are split into two parts: one for training the AI model and the other for testing or evaluation. This splitting helps to avoid overfitting, which occurs when an AI model performs exceptionally well on its training data but fails to generalize to new, unseen data. By holding out the testing (evaluation) data during training, the model’s performance is evaluated using this unseen test data, which serves as a representative sample of real-world data. In the context of computational pathology, the dataset could include histopathology images, pathology reports, immunohistochemistry images, and more, depending on the specific task the AI is designed to perform.

A public dataset (also known as a challenge or benchmark dataset) is a collection of data that is freely accessible to researchers for non-commercial use. These datasets are curated for community competitions and model evaluation, offering standardized platforms for comparison across algorithms. Public medical imaging datasets are collected or supported by universities and government agencies. Several platforms, such as Grand Challenge and Kaggle, were created for sharing biomedical imaging data that facilitates the end-to-end development of machine learning solutions through challenges and competitions. For instance, the Grand Challenge hosts several challenges in the field of computational pathology, such as PUMA, PANDA [54,55], Leopard [56], and Monkey [57]. These challenges, organized by the Department of Pathology at Radboudumc, the Netherlands, focus on developing machine learning methods for various tasks, including nuclei and tissue segmentation in melanoma histopathology, Gleason grading in PCa, predicting biochemical PCa recurrence, and histopathological assessment of transplant kidney biopsies [54,55,56,57]. These projects have been instrumental in advancing computer vision and deep learning research in cancer diagnosis using histopathology slides.

Another important public dataset is the “ImageNet project,” which is a large visual database designed to advance computer vision research. It contains over 14 million hand-annotated images across more than 20,000 categories, including animals, plants, vehicles, and more [58]. The most widely used subset of ImageNet is the ImageNet Large Scale Visual Recognition Challenge (ILSVRC) 2012–2017 image classification and localization dataset, which includes 1000 object classes and over one million images.

## 6. Conclusions

In conclusion, this review provides a foundational bridge between the fields of pathology and artificial intelligence by offering clear, accessible explanations of core AI concepts, tools, and terminology relevant to digital pathology. By focusing on practical applications, model training frameworks, image analysis nomenclature, and available public datasets, the manuscript equips pathologists with the knowledge needed to engage more effectively in interdisciplinary collaborations. Enhancing mutual understanding between pathologists and AI researchers is essential for realizing the full potential of AI-assisted diagnostics in clinical practice. This review ultimately aims to facilitate that collaboration and support the continued integration of AI into pathology for improved patient care.

## Figures and Tables

**Figure 1 diagnostics-15-01699-f001:**
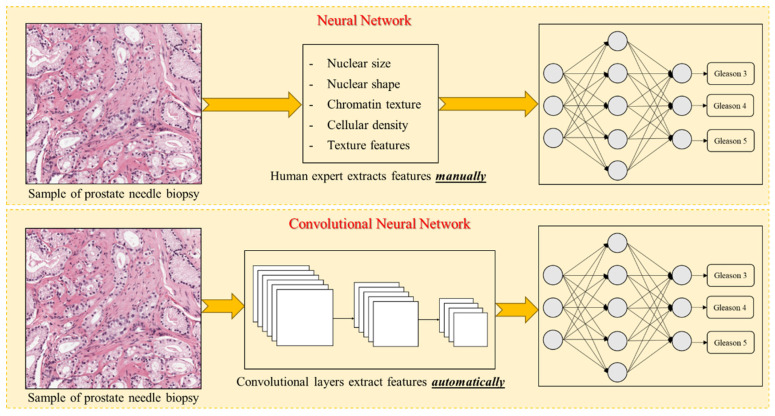
A diagram illustrates the differences between a neural network and a convolutional neural network. An image of cancerous epithelium is fed into both networks for Gleason score prediction. In the case of neural networks, a human expert must manually extract relevant features from the image for grading, whereas convolutional neural networks automatically learn these features.

**Figure 2 diagnostics-15-01699-f002:**
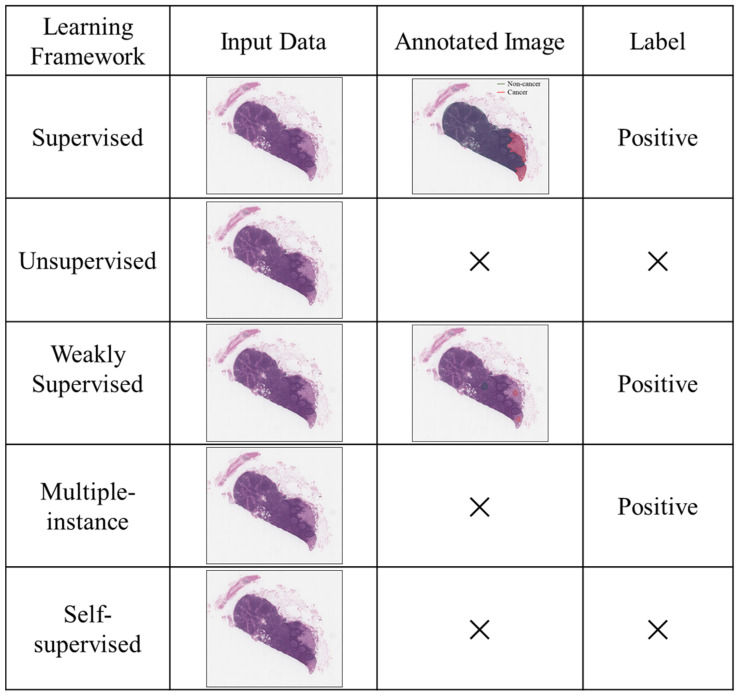
The difference between different deep learning frameworks for lymph node metastasis detection. ×: Not available.

**Figure 3 diagnostics-15-01699-f003:**
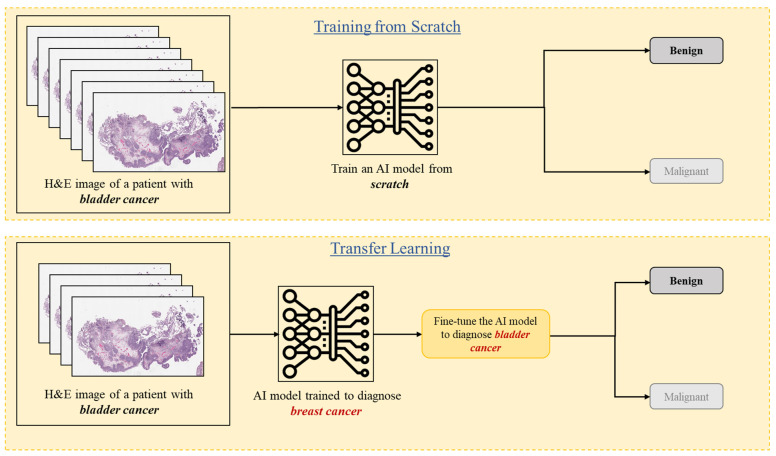
A diagram showcases the difference between transfer learning and traditional deep learning methods for detecting lymph node metastasis. In traditional deep learning, an AI model is trained from scratch using a large dataset of labeled examples. Conversely, in transfer learning, an AI model pre-trained on a specific task is fine-tuned for a different but related task using a relatively smaller labeled dataset.

**Figure 4 diagnostics-15-01699-f004:**
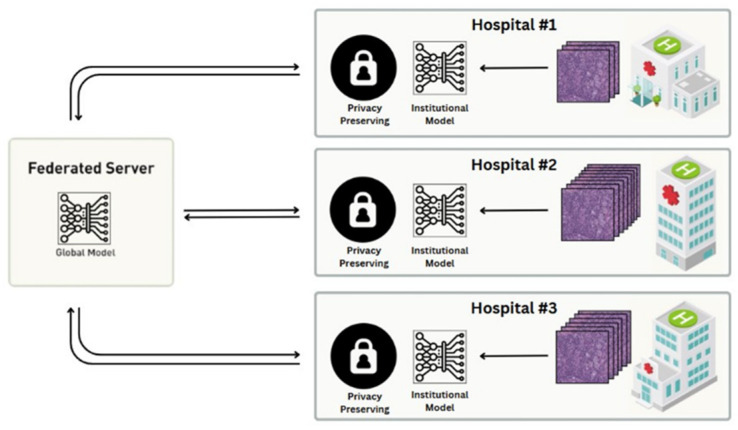
A diagram illustrates federated learning, where a trained AI model is shared among different centers, rather than centralizing data, to preserve patient privacy.

**Figure 5 diagnostics-15-01699-f005:**
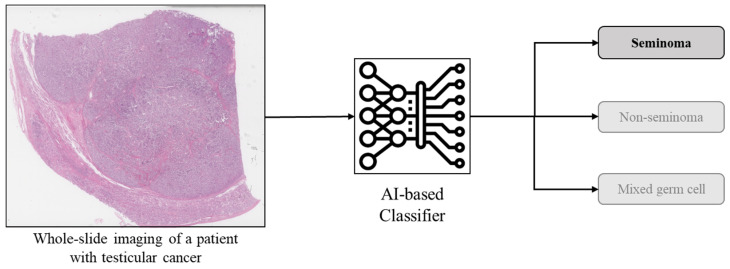
An example of AI in classification: An H&E image of a patient with testicular cancer is processed by the AI, which accurately labels it as a seminoma (vs. non-seminoma or mixed germ cell tumor).

**Figure 6 diagnostics-15-01699-f006:**
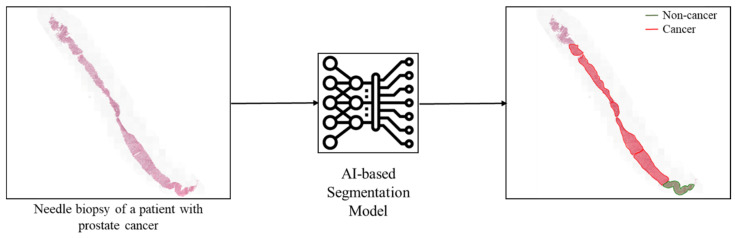
An example of AI in segmentation: An H&E image of a patient with prostate cancer is analyzed by the AI, which accurately identifies the cancer and segments the image into cancerous and non-cancerous regions.

**Table 1 diagnostics-15-01699-t001:** The scope of potential AI applications in pathology.

Application of AI in Pathology	Examples
Image analytics	Counting mitotic figures Quantifying IHC markers such as ER and PR Measuring tumor size
Disease diagnostics	Detection of prostate cancer Gleason grading for prostate cancer Breast cancer grading H. pylori detection
Outcome prediction	Predict disease prognosis Predict therapy response Patient triage for molecular testing

## Data Availability

No new data were created or analyzed in this study.

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
