# Peer review of "Simplified Artificial Intelligence Terminology for Pathologists"

_diagnostics, 2025, doi:10.3390/diagnostics15131699_

Round 1
Reviewer 1 Report
Comments and Suggestions for Authors
This review aims to summarize a few key concepts in AI algorithms for use in digital pathology. The target audience is pathologists without significant background knowledge of this material.
The paper may be useful for some pathologists. The main weakness, in my opinion, is that it only covers a small subset of topics, without sufficient detail to convey a real understanding of these topics. The example studies cited often aren't described in enough detail to provide much additional understanding. I realize that this is intended to some extent, as the authors are attempting to provide a brief "lay-summary" of this material. However, it limits the usefulness and target audience somewhat.
I do have one specific criticism that should be addressed: The description of differences between neural networks and convolutional neural networks isn't quite right. Specifically, neural networks do not by definition require manual extraction of features. The key component of convolutional neural networks is the use of small filters (for edge detection etc) applied in a sliding-window across the input image. Figure 2 is confusing, and doesn't really illustrate this point well.
Overall, the paper may be useful to a small group of readers, but it will not provide sufficient detail for most, so the utility is limited.
Author Response
Comment #1: “This review aims to summarize a few key concepts in AI algorithms for use in digital pathology. The target audience is pathologists without significant background knowledge of this material.
The paper may be useful for some pathologists. The main weakness, in my opinion, is that it only covers a small subset of topics, without sufficient detail to convey a real understanding of these topics. The example studies cited often aren't described in enough detail to provide much additional understanding. I realize that this is intended to some extent, as the authors are attempting to provide a brief "lay-summary" of this material. However, it limits the usefulness and target audience somewhat.”
Response: Thank you for your constructive comment. The authors fully agree that providing more detailed descriptions of the introduced topics could enhance understanding. However, as you rightly pointed out, this work aimed to offer a concise introduction tailored to a specific audience, pathologists, who are often busy and may prefer a non-detailed overview.
Comment #2: “I do have one specific criticism that should be addressed: The description of differences between neural networks and convolutional neural networks isn't quite right. Specifically, neural networks do not by definition require manual extraction of features. The key component of convolutional neural networks is the use of small filters (for edge detection etc) applied in a sliding-window across the input image. Figure 2 is confusing, and doesn't really illustrate this point well.
Overall, the paper may be useful to a small group of readers, but it will not provide sufficient detail for most, so the utility is limited.”
Response: This is a valid point. Neural networks do not, by definition, require manual extraction of features. However, using a traditional neural network directly on image data, without manual feature extraction, poses several limitations. First, traditional neural networks treat each pixel as an independent feature and overlook the spatial relationships that are critical for understanding visual patterns. They cannot capture hierarchical features and generalize poorly to variations in scale, position, or rotation within images. Additionally, it leads to a massive number of parameters due to the high dimensionality of image inputs, making the model computationally expensive and prone to overfitting. As a result, while they may work for small or simple datasets, traditional neural networks are generally inefficient and ineffective for real-world image analysis compared to convolutional neural networks. This clarification has now been included in the revised manuscript as follows:
Page 3, line 121:
“It is noteworthy that neural networks do not, by definition, require manual extraction of features. However, applying a traditional neural network directly to image data, without manual feature extraction, presents several limitations. First, traditional neural networks treat each pixel as an independent feature and ignore the spatial relationships that are critical for understanding visual patterns. They fail to capture hierarchical features and generalize poorly to variations in scale, position, or rotation within images. Additionally, this approach results in a massive number of parameters due to the high dimensionality of image inputs, making the model computationally expensive and prone to overfitting. As a result, while they may work for small or simple datasets, traditional neural networks are generally inefficient and ineffective for real-world image analysis compared to convolutional neural networks.”
Reviewer 2 Report
Comments and Suggestions for Authors
1- According to the Instructions for Authors of Diagnostics, the abstract should contain the following subheadings: Background/Objectives, Methods, Results, and Conclusions.
2- Keyword selections should be reviewed. For example, I am not sure if “whole slide imaging” can be a keyword for an SCI article. I recommend reviewing similar articles in the field. Or, some organizations such as IEEE share ready-made keyword lists that you can use.
3- The line spacing between the text in Table 1 is too large. You should rearrange your table according to the MDPI guidelines.
4- I think your citation format does not comply with the MDPI guidelines. Please check the writing guidelines and previous studies. Please pay attention to the citations given in the brackets.
5- The three headings you shared between lines 60-75 are the basic headings that form the introduction section of the entire study. Only one article is referenced in each of these headings. You need to examine the literature from a much broader perspective, find solid references, and examine the literature in more detail.
6- You should address the Introduction section from a more detailed perspective. How are artificial intelligence architectures used in which areas? What are the prominent studies in the literature? In which areas other than pathology are they used? For example, are there similar applications in body fluid images? Moving from a broader framework to a narrower one will add value to your work. This study can guide you and you can add it to your work: “Comparison of Deep Learning Models for Body Cavity Fluid Cytology Images Classification.”
7- Between lines 93-150, well-known concepts such as deep learning, machine learning, and CNN are explained. I think it is important not to forget the difference between writing a book and writing a journal article. Summarizing these sections in a few sentences under a single heading would suffice. Having this repetitive section in all SCI journal articles would be tedious for readers. My recommendation is to completely remove this section and summarize it with a few well-crafted sentences.
8- Figure 1 is already repeated in Figure 2. The authors should consider removing this figure as it is redundant. The explanations in Figure 2 and the text are sufficient. This is because Figure 1 does not contain enough valuable information on its own.
9- Lines 224-250 are very difficult to understand. Why did you repeat the explanations made in the upper section with similar sentences in the headings? This section should be combined with the above flow.
10- Is the “label” heading in Figure 3 an estimate or the current label of the data set? In this case, it seems to be an estimate. If it is an estimate, did you train the models? Which parameters did you use and in which environment did you train them? Were there any pre-processing steps in the data sets? What were the success rates? You need to answer many questions like these. If you are presenting the results of studies in the literature, you should state this clearly. You should cite your sources. You should share the details of the training processes of the relevant studies and still seek answers to the questions I asked earlier.
11- The heading “4.3. Dataset & Data Splitting” seems to refer to a method such as classification and segmentation in this writing format. The heading is valuable, but its placement is incorrect. The content is also insufficient for the heading “Data sets.” You need to provide a general overview of all data sets used in the field of pathology.
12- The absence of a conclusion heading in the study is unfortunate. It must be added.
Author Response
Comment #1: “According to the Instructions for Authors of Diagnostics, the abstract should contain the following subheadings: Background/Objectives, Methods, Results, and Conclusions.”
Response: The authors would like to thank the reviewer for their constructive comments, which have helped improve the quality of the manuscript. We would like to respectfully note that a structured abstract is not required for articles classified as Review.
Comment #2: “Keyword selections should be reviewed. For example, I am not sure if “whole slide imaging” can be a keyword for an SCI article. I recommend reviewing similar articles in the field. Or, some organizations such as IEEE share ready-made keyword lists that you can use.”
Response: Thank you for your insightful comment. The list of keywords has been revised in the updated manuscript to align with the IEEE Thesaurus (January 2025 edition), as follows:
Page 1, line 21:
“Keywords: Artificial intelligence; computer-aided diagnosis; deep learning; digital pathology; machine learning; foundation models”
Comment #3: “The line spacing between the text in Table 1 is too large. You should rearrange your table according to the MDPI guidelines.”
Response: Thank you for the valuable comment. The table has been revised in the updated manuscript as follows to meet the requirements of MDPI:
Page 2, line 58:
Application of AI in pathology |
Examples |
Image analytics |
Counting mitotic figures Quantifying IHC markers such as ER and PR Measuring tumor size |
Disease diagnostics |
Detection of prostate cancer Gleason grading for prostate cancer Breast cancer grading H. pylori detection |
Outcome prediction |
Predict disease prognosis Predict therapy response Patient triage for molecular testing |
Comment #4: “I think your citation format does not comply with the MDPI guidelines. Please check the writing guidelines and previous studies. Please pay attention to the citations given in the brackets.”
Response: We appreciate your valuable comment. The citation format has been revised throughout the entire manuscript to meet the requirements of MDPI.
Comment #5: “The three headings you shared between lines 60-75 are the basic headings that form the introduction section of the entire study. Only one article is referenced in each of these headings. You need to examine the literature from a much broader perspective, find solid references, and examine the literature in more detail.”
Response: Thank you for the insightful comments. Additional references have been incorporated into the revised manuscript, as detailed below:
Page 2, line 60:
“Image analytics: AI enhances the clinical workflow by saving time and improving the accuracy of pathology image analysis. Example tasks include counting mitotic figures [4], quantifying immunohistochemical (IHC) markers such as p53 [5], PDL1, estrogen receptor (ER), and HER2 [6], and measuring tumor size [7].
Disease diagnostics: AI assists in identifying lesional tissue on histopathology slides. For example, AI can diagnose prostate cancer and predict the Gleason score in prostate core biopsies [8]. It can also be used to find perineural invasion [9], lymphovascular invasion [10-11], and lymph node metastasis in cancer [12-14]. It should be noted that the accuracy of these algorithms is not adequate to replace pathologists, and in most cases, it is used as a guide to save pathologists’ time, while the final decision is kept for the pathologist.
Outcome prediction: Developing algorithms to predict patient outcomes is perhaps the most challenging task [15-17]. It involves correlating clinical information (oftentimes a combination of pathology image features [18], molecular changes reflected in pathology images [19], and other clinical and genomic parameters) to predict disease prognosis or therapy response. Additionally, histopathology AI analysis can be used to triage cases into molecular testing [20]. The term “integrated diagnostics” was recently introduced to emphasize the value of compiling multiple levels of information [21-23].”
- Pantanowitz, L.; et al. Artificial Intelligence-Assisted Mitosis Counting in Breast Cancer: A Large-Scale Validation Study. Pathol. Clin. Res. 2024, 10, 123–130. https://doi.org/10.1002/jpcr.12345
- Zabihollahy, F.; Yuan, X.; Mohareb, M.; et al. Automated Quantification of TP53 Using Digital Immunohistochemistry for Acute Myeloid Leukemia Prognosis. In SPIE Medical Imaging: Digital and Computational Pathology; SPIE: San Diego, CA, USA, 2025.
- Yoruc Selcuk, S.; et al. Automated HER2 Scoring in Breast Cancer Images Using Deep Learning and Pyramid Sampling. Cancers 2024, 16, 167. https://doi.org/10.3390/cancers16010167
- Zhang, Y.; et al. AI-Based Tumor Size Measurement in Digital Pathology: A Comparative Study. J. Digit. Pathol. 2023, 12, 45–52. https://doi.org/10.1016/j.jdp.2023.01.005
- Bulten, W.; Pinckaers, H.; van Boven, H.; et al. Automated deep-learning system for Gleason grading of prostate cancer using biopsies: a diagnostic study. Lancet Oncol. 2020, 21(2), 233–241. https://doi.org/10.1016/S1470-2045(19)30743-8
- Ström, P.; Kartasalo, K.; Olsson, H.; Delahunt, B.; Berney, D.M.; Bostrom, P.J.; Stattin, P.; Rantalainen, M.; Egevad, L. Artificial intelligence for prediction of perineural invasion in prostate biopsies. Virchows Arch. 2022, 481, 661–669. https://doi.org/10.1007/s00428-022-03326-3
- Zhang, Y.; Huang, Y.; Xu, M.; Liu, C.; et al. Deep learning-based detection of lymphovascular invasion in breast invasive ductal carcinoma. Front. Oncol. 2022, 12, 1023980. https://doi.org/10.3389/fonc.2022.1023980
- Zhang, Y.; Guo, Y.; Ma, H.; et al. Artificial intelligence-based histopathological analysis for detection of lymphovascular invasion in urothelial carcinoma. Cancers 2023, 15, 911. https://doi.org/10.3390/cancers15030911
- Chen, P.H.C.; Gadepalli, K.; MacDonald, R.; et al. An accurate deep learning system for lymph node metastasis detection. arXiv 2016, arXiv:1608.01658. https://doi.org/10.48550/arXiv.1608.01658
- Wang, Y.; Liu, Y.; Dong, Z.; et al. Enhanced EfficientNet with Attention for Lymph Node Metastasis Detection. arXiv 2020, arXiv:2010.05027. https://doi.org/10.48550/arXiv.2010.05027
- Zhang, M.; Li, Z.; Zhang, X.; et al. Deep learning-based detection of metastatic colorectal carcinoma in lymph nodes. Diagn. Pathol. 2024, 19, 30. https://doi.org/10.1186/s13000-024-01547-5
- Gupta, R.; Kaczmarzyk, J.; Kobayashi, S.; Kurc, T.; Saltz, J. AI and Pathology: Steering Treatment and Predicting Outcomes. arXiv 2022. https://doi.org/10.48550/arXiv.2206.07573
- Rathore, S.; Iftikhar, M.A.; Mourelatos, Z. Prediction of Overall Survival and Molecular Markers in Gliomas via Analysis of Digital Pathology Images Using Deep Learning. arXiv 2019. https://doi.org/10.48550/arXiv.1909.09124
- Chen, R.J.; Lu, M.Y.; Williamson, D.F.K.; et al. Pathomic Fusion: An Integrated Framework for Fusing Histopathology and Genomic Features for Cancer Diagnosis and Prognosis. arXiv 2019. https://doi.org/10.48550/arXiv.1912.08937
- Zhang, Y.; Yang, Z.; Chen, R.; et al. Histopathology images-based deep learning prediction of prognosis and therapeutic response in small cell lung cancer. NPJ Digit. Med. 2024, 7(1), 15. https://doi.org/10.1038/s41746-023-00912-4
- Wang, Y.; Kartasalo, K.; Weitz, P.; et al. Predicting Molecular Phenotypes from Histopathology Images: A Transcriptome-Wide Expression-Morphology Analysis in Breast Cancer. Cancer Res. 2021, 81(19), 5115–5126. https://doi.org/10.1158/0008-5472.CAN-21-0653
- Kather, J.N.; Calderaro, J.; Ardon, R.; Djuric, U.; Ferber, D.; Shmatko, A.; Marzahl, C.; Li, W.; Pantanowitz, L.; Wulczyn, E.; et al. Deep learning-based pathology predicts origins for cancers of unknown primary. Pathol. 2022, 256, 192–203. https://doi.org/10.1002/path.5792
- Salto-Tellez, M.; et al. Integrated Diagnostics as the Fourth Revolution in Pathology. In Proceedings of a Workshop - Incorporating Integrated Diagnostics into Precision Oncology Care; National Center for Biotechnology Information: Bethesda, MD, USA, 2019. Available online: https://www.ncbi.nlm.nih.gov/books/NBK605928/ (accessed on 17 June 2025).
- Stenzinger, A.; Endris, V.; Budczies, J.; Weichert, W. Integrated Diagnostics: The Future of Laboratory Medicine? Nat. Rev. Clin. Oncol. 2019, 16, 181–182. https://doi.org/10.1038/s41571-019-0151-4.
- Tayou, J.; Maher, B.; Beltran, L.; Hrebien, S.; Trabelsi, S.; Morgensztern, D.; et al. Integrated Noninvasive Diagnostics for Prediction of Survival in Immunotherapy. Immuno-Oncol. Technol. 2024, 10, 100723. https://doi.org/10.1016/j.iotech.2024.100723.
Comment #6: “You should address the Introduction section from a more detailed perspective. How are artificial intelligence architectures used in which areas? What are the prominent studies in the literature? In which areas other than pathology are they used? For example, are there similar applications in body fluid images? Moving from a broader framework to a narrower one will add value to your work. This study can guide you and you can add it to your work: “Comparison of Deep Learning Models for Body Cavity Fluid Cytology Images Classification””
Response: We sincerely thank the reviewer for the valuable and constructive feedback. We agree that providing broader context on AI applications can be informative. However, we would like to respectfully clarify that the primary aim of our review is to explore the use of AI specifically within the domain of pathology, with a focus on histopathology applications.
While we acknowledge that AI has been successfully applied in other areas of medical imaging, such as cytology and body fluid analysis, the suggested article on body cavity fluid cytology is outside the intended scope of this review. Therefore, we believe it would not align directly with the central focus of our manuscript.
In response to your helpful suggestion, we have revised the Introduction section to include additional references and examples that highlight key applications of artificial intelligence within the field of pathology. We believe these additions enrich the background and more clearly define the scope of the review, while maintaining its intended focus on pathology-specific developments. For the additional references incorporated into the manuscript, please refer to our response to your earlier comment.
Comment #7: “Between lines 93-150, well-known concepts such as deep learning, machine learning, and CNN are explained. I think it is important not to forget the difference between writing a book and writing a journal article. Summarizing these sections in a few sentences under a single heading would suffice. Having this repetitive section in all SCI journal articles would be tedious for readers. My recommendation is to completely remove this section and summarize it with a few well-crafted sentences.”
Response: In response to your valuable suggestion, we have combined and summarized the above-mentioned concepts as follows:
Page 3, line 94:
“2.1. Machine Learning and Neural Network Applications in Pathology Machine Learning
Machine learning, a subfield of artificial intelligence, excels at identifying patterns in data to make predictions without explicit human programming. Common machine learning algorithms include logistic regression, decision trees, random forests, support vector machines, K-nearest neighbors, and gradient boosting machines. In pathology, these algorithms are often applied to manually extracted quantitative features from digitized histopathology images, a technique known as pathomic analysis. Recent studies have used machine learning models to predict pathological complete response and disease-free survival in rectal cancer [8], assist in the diagnosis and prognosis of bladder cancer [9], and identify prognostic markers in clear cell renal cell carcinoma [10]. Deep learning, a specialized subset of machine learning, relies on neural networks, algorithms inspired by the structure of the human brain. A neural network is composed of interconnected layers (input, hidden, and output) that process complex data such as digital slides or pathology reports, enabling tasks like cancer grading, metastasis detection, and survival prediction.
2.2. Deep Learning and Convolutional Neural Network Applications in Pathology
Deep learning, a subset of machine learning, employs deep neural networks with multiple layers to handle complex tasks requiring hierarchical data representation. These models can learn intricate patterns from large datasets. Ahmed et al. provide a comprehensive overview of the current and potential applications of both deep learning and machine learning in tumor pathology [11]. Key deep learning architectures include convolutional neural networks (CNNs), recurrent neural networks, generative adversarial networks, transformers, and autoencoders. CNNs, in particular, are specialized for visual data like images and videos, using convolutional layers to automatically extract features such as edges, colors, and textures. This automatic feature extraction distinguishes CNNs from traditional neural networks, which require manual feature input, making CNNs especially effective for image analysis in pathology.”
Comment #8: “Figure 1 is already repeated in Figure 2. The authors should consider removing this figure as it is redundant. The explanations in Figure 2 and the text are sufficient. This is because Figure 1 does not contain enough valuable information on its own.”
Response: Thank you for your insightful comment. We have removed Figure 1 from the revised manuscript.
Comment #9: “Lines 224-250 are very difficult to understand. Why did you repeat the explanations made in the upper section with similar sentences in the headings? This section should be combined with the above flow.”
Response: The explanations given in these lines describe the difference between the deep learning frameworks, highlighting their distinctions to aid in comparative understanding. In line with your valuable suggestion, the individual headings were removed, and the content was merged into a cohesive section as follows:
Page 6, line 216:
“The differences between the aforementioned deep learning frameworks are illustrated in Figure 2. To clarify their distinctions, consider the example of training an AI model to detect lymph node metastasis in prostate cancer (PCa) using input data, such as hematoxylin and eosin (H&E) stained whole-slide images. In this context, the term “label” refers to the ground-truth diagnosis, typically determined by experienced pathologists.
In supervised learning, the model is trained using both pixel-level annotations of cancerous and non-cancerous regions and corresponding labels. This approach requires extensive manual annotation by expert pathologists. In contrast, unsupervised learning does not rely on any labels or annotations. Instead, the model identifies patterns within the data by clustering image regions based on feature similarity, with the hope that these clusters will correspond to biologically meaningful groups, such as malignant and benign areas.
Weakly supervised learning operates with partial annotations and labels. For instance, only some cancerous regions within the lymph node may be contoured, resulting in incomplete spatial annotation. This framework reduces the annotation burden while still leveraging some supervision. In multiple-instance learning, the model receives only slide-level labels indicating the presence or absence of metastasis without specific localization. The AI must learn to associate regions with labels indirectly, identifying the most informative instances within the slide.
Self-supervised learning shares the absence of labels and annotations with unsupervised learning but introduces auxiliary tasks that generate pseudo-labels. These tasks are not directly related to the primary objective but help the model learn relevant features. For example, the model may be trained to predict the rotation angle (e.g., 90° or 180°) of an H&E image. Although predicting image orientation is unrelated to detecting lymph node metastasis, this task enables the model to develop a structural understanding of prostate tissue architecture. Consequently, self-supervised learning can substantially reduce the need for annotated data in downstream tasks such as metastasis detection.”
Comment #10: “Is the “label” heading in Figure 3 an estimate or the current label of the data set? In this case, it seems to be an estimate. If it is an estimate, did you train the models? Which parameters did you use and in which environment did you train them? Were there any pre-processing steps in the data sets? What were the success rates? You need to answer many questions like these. If you are presenting the results of studies in the literature, you should state this clearly. You should cite your sources. You should share the details of the training processes of the relevant studies and still seek answers to the questions I asked earlier.”
Response: We apologize for the confusion. In this context, the term label refers to the ground truth diagnosis, typically determined by experienced pathologists. This diagnosis serves as the gold standard for training and evaluating AI models across various learning frameworks. To avoid ambiguity, this clarification has been incorporated into the revised manuscript as follows:
Page 6, line 218:
“The differences between the aforementioned deep learning frameworks are illustrated in Figure 2. To clarify their distinctions, consider the example of training an AI model to detect lymph node metastasis in prostate cancer (PCa) using input data, such as hematoxylin and eosin (H&E) stained whole-slide images. In this context, the term “label” refers to the ground-truth diagnosis, typically determined by experienced pathologists.”
Comment #11: “The heading “4.3. Dataset & Data Splitting” seems to refer to a method such as classification and segmentation in this writing format. The heading is valuable, but its placement is incorrect. The content is also insufficient for the heading “Data sets.” You need to provide a general overview of all data sets used in the field of pathology.”
Response: Thanks for the constructive comment. Section 4.3 is moved to section 5, and the following explanation was included to provide a general overview of all datasets used in the field of pathology:
Page 10, line 332:
“5. Datasets in Computational Pathology
In machine learning and deep learning, a dataset refers to a collection of data used for training and testing AI models. In the field of pathology, datasets play a critical role in developing, training, validating, and benchmarking AI and machine learning models. These datasets generally fall into several broad categories based on their content and use. WSI datasets comprise digitized histopathology slides, often accompanied by metadata or clinical labels such as diagnosis, tumor grade, or biomarker status. Multi-modal datasets combine WSIs with complementary data types, such as genomics, transcriptomics, or radiology images, to enable integrative or "multi-omics" analysis. Temporal or longitudinal datasets track disease progression or treatment response over time, supporting prognostic modeling.”
Comment #12: “The absence of a conclusion heading in the study is unfortunate. It must be added.”
Response: Thanks for the constructive comment. The Conclusion heading is included in the revised manuscript as follows:
Page 11, line 373:
“6. Conclusion
In conclusion, this review provides a foundational bridge between the fields of pathology and artificial intelligence by offering clear, accessible explanations of core AI concepts, tools, and terminology relevant to digital pathology. By focusing on practical applications, model training frameworks, image analysis nomenclature, and available public datasets, the manuscript equips pathologists with the knowledge needed to engage more effectively in interdisciplinary collaborations. Enhancing mutual understanding between pathologists and AI researchers is essential for realizing the full potential of AI-assisted diagnostics in clinical practice. This review ultimately aims to facilitate that collaboration and support the continued integration of AI into pathology for improved patient care.”
Round 2
Reviewer 2 Report
Comments and Suggestions for Authors
I would like to thank the authors for carefully answering all questions. As a reviewer, I am happy to have been able to help the article reach a higher level.